# Biotic Cathode of Graphite Fibre Brush for Improved Application in Microbial Fuel Cells

**DOI:** 10.3390/molecules27031045

**Published:** 2022-02-03

**Authors:** Siti Farah Nadiah Rusli, Siti Mariam Daud, Mimi Hani Abu Bakar, Kee Shyuan Loh, Mohd Shahbudin Masdar

**Affiliations:** 1Fuel Cell Institute, Universiti Kebangsaan Malaysia, Bandar Baru Bangi 43600, Malaysia; sitifarahnadiahrusli@gmail.com (S.F.N.R.); ksloh@ukm.edu.my (K.S.L.); shahbud@ukm.edu.my (M.S.M.); 2Human Genome Centre, School of Medical Sciences, Universiti Sains Malaysia, Kota Bharu 16150, Malaysia; yummiemariam@gmail.com; 3Department of Chemical & Process Engineering, Faculty of Engineering and Built Environment, Universiti Kebangsaan Malaysia, Bandar Baru Bangi 43600, Malaysia

**Keywords:** microbial electrochemical technology, graphite fibre brush, biocathode, electron transfer

## Abstract

The biocathode in a microbial fuel cell (MFC) system is a promising and a cheap alternative method to improve cathode reaction performance. This study aims to identify the effect of the electrode combination between non-chemical modified stainless steel (SS) and graphite fibre brush (GFB) for constructing bio-electrodes in an MFC. In this study, the MFC had two chambers, separated by a cation exchange membrane, and underwent a total of four different treatments with different electrode arrangements (anodeǁcathode)—SSǁSS (control), GFBǁSS, GFBǁGFB and SSǁGFB. Both electrodes were heat-treated to improve surface oxidation. On the 20th day of the operation, the GFBǁGFB arrangement generated the highest power density, up to 3.03 W/m^3^ (177 A/m^3^), followed by the SSǁGFB (0.0106 W/m^3^, 0.412 A/m^3^), the GFBǁSS (0.0283 W/m^3^, 17.1 A/m^3^), and the SSǁSS arrangements (0.0069 W/m^−3^, 1.64 A/m^3^). The GFBǁGFB had the lowest internal resistance (0.2 kΩ), corresponding to the highest power output. The other electrode arrangements, SSǁGFB, GFBǁSS, and SSǁSS, showed very high internal resistance (82 kΩ, 2.1 kΩ and 18 kΩ, respectively) due to the low proton and electron movement activity in the MFC systems. The results show that GFB materials can be used as anode and cathode in a fully biotic MFC system.

## 1. Introduction

Microbial fuel cells (MFCs) are environmentally friendly devices where carbonaceous materials become the biomass for electricity generation. MFCs consist of anode and cathode chambers, which are divided by a separator [1,2]. Transferring electrons from the anode to the cathode generates electrical power and reduces organic waste. This type of fuel cell shows a double benefit over other devices since energy production and treatment coincide. The use of MFCs for electricity generation with wastewater has already been widely reported in the literature [3,4].

However, there are several issues hindering the commercialisation of this technology. The primary concerns involve MFC design optimisation to maximise power output and the high cost of installation and operation [5,6]. An MFC uses microorganisms as biocatalysts at the anode, while platinum is usually used as the catalyst for the cathode. The application of the platinum catalyst adds a significant limitation to MFC application and economic viability. Compared to platinum, a biocathode is a promising way to improve the cathode reaction performance without additional investment or risk of pollution [7]. Various materials, including carbon-based materials and stainless steel, have been tested for their suitability as biocathodes in MFCs. Zhang et al. conducted a comparison study of materials for an MFC biocathode including graphite felt, stainless steel (SS) mesh and carbon paper [8]. The results showed that graphite felt, which is thick and has a loose texture, produced the highest current density and power (350 mA/m^2^ and 109.5 mW/m^2^) compared to carbon paper (210 mA/m^2^ and 32.7 mW/m^2^) and SS mesh (18 mA/m^2^ and 3.1 mW/m^2^) [8,9]. Another study by Kargi et al. [10] reported the presence of a biofilm on the SS cathode surface in their MFC under seawater conditions. The biofilm proves to be an efficient catalyst for oxygen reduction while generating a current density of 189 mA/m^2^.

A current density of up to 220 mA/m^2^, normalised to the cathode projected surface area, was obtained when oxygen became the electron acceptor on a graphite felt or woven carbon fibre-based biocathode [11]. You et al. (2009) carried out a study on biocathode performance to compare several carbonaceous materials, such as graphite fibre brush (GFB), under biotic and abiotic conditions, in identical MFC systems [12]. Their results showed that MFCs with biocathodes obtained the highest power (68.4 W/m^3^) compared to those with an abiotic system (31.5 W/m^3^). The improvements achieved from the use of a biocathode are effective and more sustainable in the MFC system, thereby making it viable as an alternative on issues concerning chemical catalysts.

The materials and design are among the most critical challenges in building a cost-effective MFC system for bio-electrodes. MFC studies on biocathodes are lacking in terms of materials and pairing of electrode materials compared to those of bioanodes. Based on previous biocathode studies, GFB and SS often give good results as supporting material, such as good power generation and wide surface area for bacterial attachment.

In this study, we reported the observation on the effect of combined materials between the GFB and the SS electrodes and the non-chemical surface modification of GFB in MFC application.

## 2. Results

### 2.1. Power and Polarisation Output

All systems with different electrode pairing arrangements started with an abiotic cathode with ferricyanide catholyte to strengthen the anodic biofilm, before fully operating in biotic mode on day 60.

#### 2.1.1. Abiotic Cathode for 60 Days

The electrochemical analyses were performed on day 20 with ferricyanide as the catholyte. The measured open circuit voltage (OCV) of the GFBǁGFB system was recorded at 0.56 V (Figure 1A), which was lower than that of SSǁSS (0.64 V) and GFBǁSS (0.58 V) by 13% and 19%, respectively. However, the OCV of the GFBǁGFB system was slightly higher (by 3%) than that of SSǁGFB (0.47 V). At closed circuit, the GFBǁGFB system generated the highest maximum volumetric power density, up to 3.03 ± 0.7 W/m^3^ (177 A/m^3^) (Figure 1B), which was higher than that of other systems in this study (Figure 1B, expanded scales)—99% from GFBǁSS (0.0283 ± 0.002 W/m^3^, 17.1 A/m^3^), 99.6% from SSǁGFB (0.0106 ± 0.001 W/m^3^, 0.412 A/m^3^) and 99.8% from SSǁSS (0.0069 ± 0.002 W/m^3^, 1.64 A/m^3^). The high power of the GFBǁGFB system showed that the system suffered minimal ohmic loss with a calculated Rint of 0.2 kΩ (Figure 2A). Other systems with different electrode arrangements had ohmic losses and Rint nearing 100% more than that of the GFBǁGFB system (Figure 2B, expanded scale)—SSǁGFB (82 kΩ) > SSǁSS (18 kΩ) > GFBǁSS (2.1 kΩ).

#### 2.1.2. Biotic Cathode

Electrochemical analyses were performed 20 days after the systems transformed into being fully biotic. The measured open circuit voltage (OCV) of the GFBǁGFB system was recorded at 0.33 V (Figure 2A), which was lower than that of SSǁGFB (0.74 V) and of GFBǁSS (0.50 V) by 55% and 34%, respectively (Figure 2B). However, the OCV of the GFBǁGFB system was slightly higher (by 12%) than that of SSǁSS (0.29 V). The highest maximum volumetric power density was that of the GFBǁGFB system (Figure 2C) (0.597 ± 0.13 W/m^3^), followed by the rest (Figure 2C,D, expanded scales)—67% more than that of SSǁGFB (0.196 ± 0.01 W/m^3^), 99.5% more than that of GFBǁSS (0.003 ± 0.0003 W/m^3^) and 100% more than that of SSǁSS (0.0004 ± 0.0009 W/m^3^). Calculations from Figure 2A,B show that GFBǁGFB had a Rint value of 0.2 kΩ, with minimal ohmic losses, which is almost 100% lower than those of other systems in this study: SSǁGFB (3 kΩ) < GFBǁSS (71 kΩ) < SSǁSS (157 kΩ).

### 2.2. Chemical Oxygen Requirements

Carbonaceous substrate indirectly provides the electrons needed to operate the metabolic process in the microorganism and generates power. A chemical oxygen demand (COD) reduction represents an efficient functional mixed culture in wastewater treatment [13]. Other researchers have shown the relative compatibility between most current generations and COD removals [9,13,14,15]. This study showed that the COD reduction activity was at its lowest before day 40 and reached its highest stability soon afterwards. On calculating the average of the five days recorded in this study, GFBǁGFB and SSǁGFB showed the highest COD removal of up to 85%, followed by GFBǁSS (76%) and SSǁSS (75%) (Figure 3A). The COD removal activity recorded before day 40, however, was inversed with the generated maximum power density (Figure 3B).

### 2.3. Morphology Biofilm on Electrodes

Surface modification during the fabrication of an electrode can improve bacterial adhesion and electron transfer [16,17]. The SEM images in Figure 4B,D show that bacterial cells had colonised the GFB and SS after 100 days compared to the clean electrodes (Figure 4A,C).

## 3. Discussion

### 3.1. Electrochemical Performance

The polarisation curves recorded from the abiotic cathode in the MFC on day 20 show that the SSǁGFB system experienced high overpotential losses, particularly on the anode side. This overpotential may be due to obstacles in the mass transfer of active species to the SS anode electrode, which caused rate limitation in the system (Figure 1B). Usually, the electroactive bacteria become the electron donor for an anode electrode, which requires strong communication between the electrode and the bacteria. Liu et al. reported that their heat-treated SS felt MFCs gave a faster start-up than the non-treated SS felt and graphite felt due to the formation of 3D iron–oxide nanoparticles [18]. The idea of the 3D nanostructure was suggested by Guo et at., who heated SS felt at 600 °C for 5 min, which created the 3D iron–oxide nanoparticles that enhanced the biocompatibility of the SS [19]. However, the low performance of SS in this study may be due to the smooth surface, intrinsic of the SS mesh applied compared to the SS felt that hinders microbial attachment. Thus, further modification is necessary for the smooth SS to induce more bacterial patching on its surface [20].

Meanwhile, the GFB electrode provided a higher surface area and porosity than the SS mesh [21]. These physical properties facilitate the attachment of bacteria to the electrode surface and increase the rate of electron conversion [22]. A high surface area allows the accommodation of more bacteria, which reduces resistance in charge transfer and bio-chemical reactions, thus generating more power [23], which agrees with the findings of this study (Figure 1). The result obtained from the power and polarisation curves in this abiotic MFC study shows that the electroactive biofilm attachment is critical to the anode. Hence, the material used as anode should be conductive, with a coarse surface to increase surface area. Power production comparison between the biotic and abiotic cathode systems (Figure 1 and Figure 2) in this study reveals that the selection of cathode material plays a more critical role in biotic conditions than that of the anode material. As discussed in the abiotic section, the GFB had a higher surface area than the SS, thus attracting more microbes to the cathode and producing higher electricity than MFCs with SS cathodes.

The application of the ferricyanide in the catholyte, however, managed to reduce the Rint of most MFCs in this study, except for SSǁGFB. The SSǁGFB system had the Rint reduced up to 96% (3 kΩ) when transformed to a biotic system with the maximum power density increased up to 95% (0.196 W/m^3^). Although ferricyanide has high electron affinity and can give higher power to an MFC system [24], the application of the GFB as the cathode material managed to attract the biofilm attachment and reduce the Rint of the system. For instance, the GFBǁGFB system in this study could generate a maximum power density of more than 80% when in the abiotic condition. The high generated power in the MFC from the abiotic GFBǁGFB system is almost as achieved by Jiang et al. with the maximum power density of 8.5 W/m^3^ and effective anodic volume of 320 mL [24] that is three-fold more than that found in this study. However, the biocathodes provide an inexpensive solution, without the need for an external mediator, and simple improvements in design can reduce the MFC Rint and enhance power production. For instance, a biotic MFC system can generate up to 4 W/m^3^ by increasing the electrode surface area using graphite granules for both electrodes [25]. Packing small graphite felts together as an electrode, as Cao et al. demonstrated, also increased the active surface area of electrodes and generated a maximum power density up to 26.2 W/m^3^ (71.8 A/m^3^) [26]. Rusli et al. [7] reported that GFB anode surface modification with aryl diazonium increased current generation up to 6% compared to the unmodified GFB. The modification provided positively charged groups to the anode surface, which attracted the negative charges of exoelectrogenic bacteria onto the surface [27]. Table 1 shows the importance of having electrodes with a high surface area to promote power generation in the system. Nevertheless, a long-continuous operation of a fully bioelectrode MFC may result in much poorer performance compared to a partial bioelectrode MFC, especially when a selective membrane, such as a cation exchange membrane (CEM), is used as a separator. The growth of microbes may restrict the movement of selective ions across the separator, which later causes an imbalance in system pH and electroneutrality [28] due to biofouling [29].

### 3.2. COD Reduction

The high COD reduction while generating a low current, recorded from other electrode pairs in this study (Figure 5), seems to agree with Lee et al. who used molasses wastewater as a carbon source [13]. They reported that their single-chambered MFC recorded higher COD reduction (90%) compared to the double-chambered MFC (50%), while the double-chambered MFC generated a power density that was 2.2 times higher than that produced by the single-chamber. These results show that some power generation might not be concurrent with the COD removal. This occurrence might be linked to non-exoelectrogenic microbes, which do not generate power, especially while using mixed culture as inoculum [36]. However, the existence of these non-exoelectrogenic microbes in the anode chamber harms the power performance of the MFC. Although these microbes assist in reducing the COD from the organic contaminant, they also limit the food supply to the exoelectrogenic microbes responsible for electricity generation.

### 3.3. Surface Morphology

A previous study reported that the quality of the electrode surface, such as smoothness or roughness (Figure 4A,C), affected the current density by encouraging different microbe colonisation, as shown in Figure 4B,D [21]. A biofilm formation happened when cells started growing in extracellular polymeric substances (EPS). Although the biofilm seemed dense on the SS surface (Figure 4D), the current production was deficient. The SS surface has a layer of oxides that attracts the adherence of microbes; however, there is also chromium. The heat treatment method in this study could be inadequate to reduce the composition of chromium on the SS surface [20]. As a result, the SS surface could be surrounded by inactive bacteria due to chromium poisoning, thus inhibiting power production.

## 4. Materials and Methods

### 4.1. Preparation of Microbial Fuel Cell Reactors

Two-chambered MFCs were made of acrylic material with dimensions of 70 mm × 70 mm × 20 mm, and the chambers were separated by a CEM (CEM; CMI-7000, Membrane International Inc., Glen Rock, NJ, USA). The electrodes used were made of SS mesh (from a local hardware store) with a dimension of 50 mm × 50 mm, which was made as control, and GFB (Carbon fibre tow, Zoltek Panax 35 50K, St. Louis, MO, USA). The GFB electrode was pre-assembled by pinching the fibres with dimensions of 50 mm × 20 mm into twisted titanium wire. The MFCs were subjected to four different treatments with replications of two for each treatment.

The treatment focused on the electrode pairing arrangement in the MFCs—SSǁSS (control), GFBǁSS, GFBǁGFB and SSǁGFB. The SS was made as control due to its well-known conductive nature. The GFB was pre-assembled by pinching carbon fibres (50 mm × 20 mm) into twisted titanium wire following the method used in a previous study [37]. Later, GFB electrodes were soaked overnight in acetone solution, treated with heat at 450 °C, and washed with distilled water [38]. SS mesh was prepared by soaking it in a 0.1 M solution of hydrochloric acid for two hours, again soaking it in a solution of 50–50% acetone for 30 min, and then washing it with deionised water.

The electrodes were treated with heat at a temperature of 600 °C for 5 min, a method adapted from Guo et al. [19] Mud was obtained from the Universiti Kebangsaan Malaysia (UKM) lake as a source of inoculum for the anode. The inoculation was within a ratio of 1:1 mud and 50 mM phosphate buffer solution (PBS) with the organic substrate (1.0 g/L acetate). PBS contains Na_2_HPO_4_, 4.58 g/L; NaH_2_PO_4_ • H_2_O, 2.45 g/L; NH_4_Cl, 0.31 g/L; KCl, 0.13 g/L; mineral, 12.5 mL/L and vitamin 5 mL/L. All anode chambers were operated in anaerobic conditions, while the cathode chambers were aerated using fish pumps. The anode and cathode electrodes were connected by a copper wire and a resistor to produce a closed circuit. These MFC systems ran at room temperature. Prior to the operation of the MFC system, a leak test was performed using water to ensure that no leakage occurred during the study.

### 4.2. Operation of the Microbial Fuel Cell Cathode

The reactor MFC arrangements are shown in Figure 5. The cathode chamber was first operated in an abiotic (no aeration) condition before it became fully biotic (with aeration supplied). Two types of electrolytes were used in the MFC cathode abiotic system, namely ferricyanide and PBS. At start-up until 20 days, ferricyanide solution was used under the external load of 1000 Ω, then replaced with PBS with reduced external load to 100 Ω, to increase current generation. PBS solution was used to replace the ferricyanide to prepare the cathode chamber for the biotic environment. All abiotic MFC systems transformed to iocathodes at the end of 60 days after the start-up. Each cathode chamber was later enriched, similar to anode inoculation, with PBS as the electrolyte, excluding the acetate.

### 4.3. Data Acquisition and Analysis

Voltage for all systems was recorded using a Keithley 2700/7700 multimeter (Model 2700, Keithley, Cleveland, OH, USA) every 5 min. The reduced voltage gain data were used in the calculation of current density. The polarisation curves were determined using the Linear Sweep Voltammetry method, with the help of a potentiostat (Autolab PGSTAT128N, Metrohm Autolab, Utrecht, The Netherlands). This analysis was carried out at a scan rate of 10mV/s between scanned potentials of −800 mV to 800 mV. The open-circuit voltage (OCV) was measured once the system was stable. Meanwhile, the internal resistance (Rint) was calculated from the slope of the polarisation curve. Electrode image analysis from the Electron Scanner Microscope (Zeiss, Supra 55VP, Oberkochen, Germany) was performed to study the morphology of the electrode under 200× magnification. Some calculations were normalised to the anolyte volume of the system.

## 5. Conclusions

The combination of GFB as anode and cathode electrodes generated the highest power and the lowest Rint in both the abiotic and biotic MFCs. In the case of the biotic MFC, the GFBǁGFB system generated the highest power density, up to 0.523 W/m^3^ (102 A/m^3^, 0.2 kΩ), whilst SSǁSS generated the lowest power density of 0.0004 W/m^3^ (0.09 A/m^3^, 157 kΩ). Most of the abiotic MFC shows a similar, although higher, electrochemical trend to the biotic MFC. The high electrochemical performance could be due to less resistance from the ferricyanide catholyte. COD removal was more than 90%, potentially due to the exoelectrogenic and non-exoelectrogenic microbes in the anodic chamber. The GFB has a rougher surface than the SS, which attracted different colonisation of microbes to the electrode surface and produced a high-power density.

## Figures and Tables

**Figure 1 molecules-27-01045-f001:**
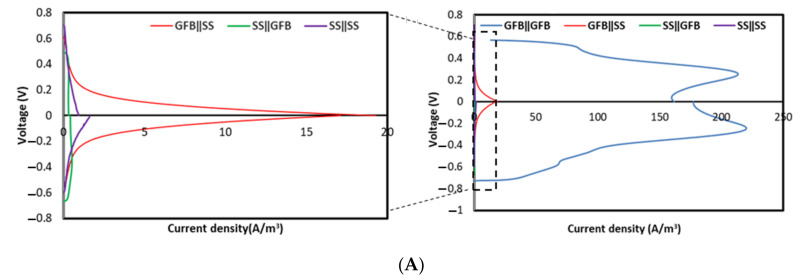
(**A**) Polarisation and (**B**) power curves at day 20 for abiotic cathode systems using ferricyanide as catholyte. The respective superpositions are located on the left side of each figure. (Note: all data are representative of the best performed unit within its replication set.).

**Figure 2 molecules-27-01045-f002:**
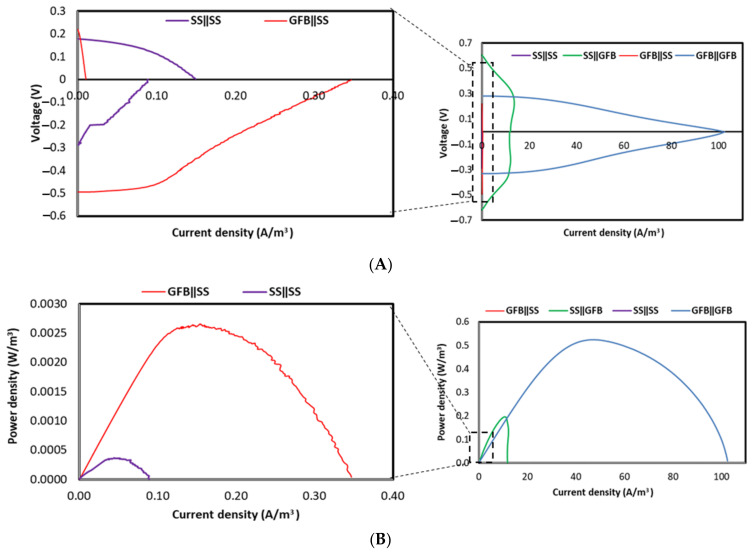
(**A**) Polarisation and (**B**) power curves at day 80 for biotic cathode systems using PBS as catholyte. The respective superpositions are located on the left side of each figure. (Note: all data are representative of the best performed unit within its replication set.).

**Figure 3 molecules-27-01045-f003:**
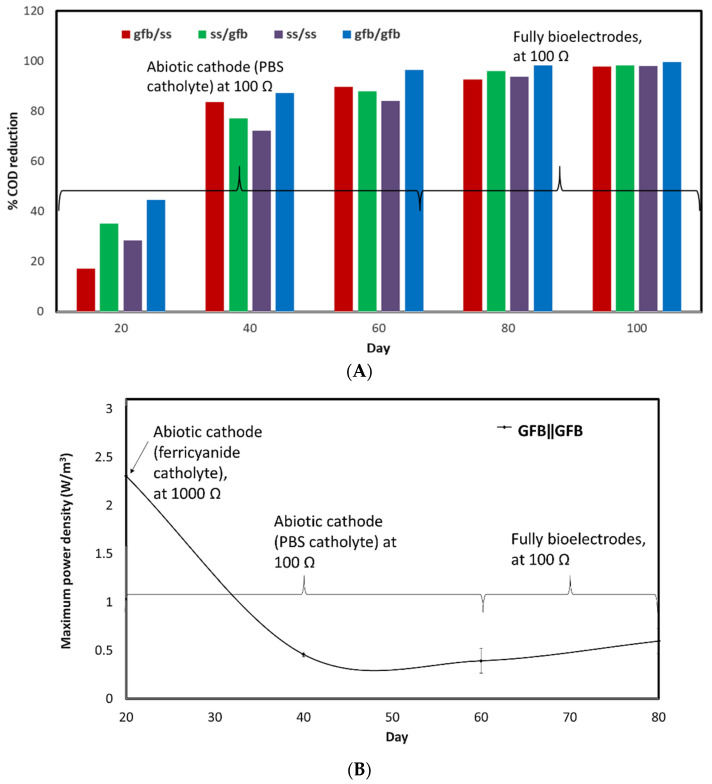
(**A**) COD reduction for all systems and (**B**) the maximum power density obtained by the GFBǁGFB system, from day 20 to day 80.

**Figure 4 molecules-27-01045-f004:**
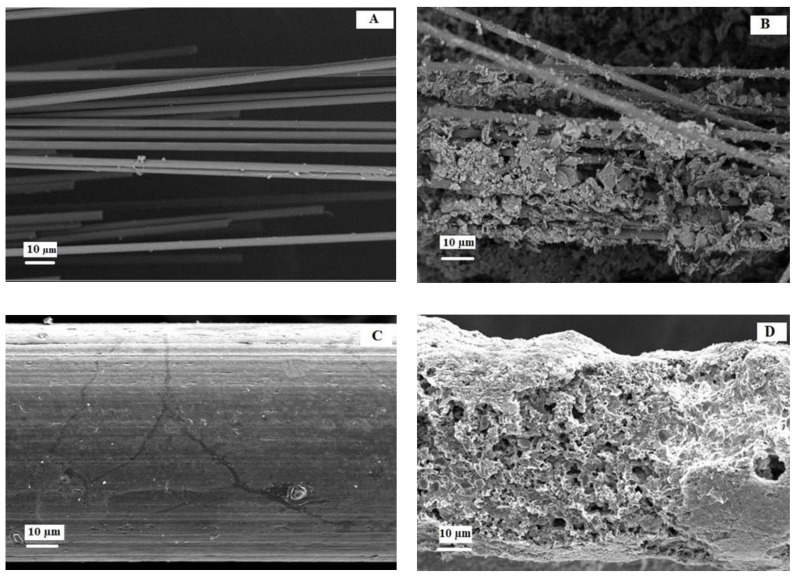
SEM images of (**A**) clean graphite brush; (**B**) graphite brush with a biofilm on the cathode; (**C**) clean stainless steel; and (**D**) stainless steel with a biofilm on the cathode. (Magnification: 200×).

**Figure 5 molecules-27-01045-f005:**
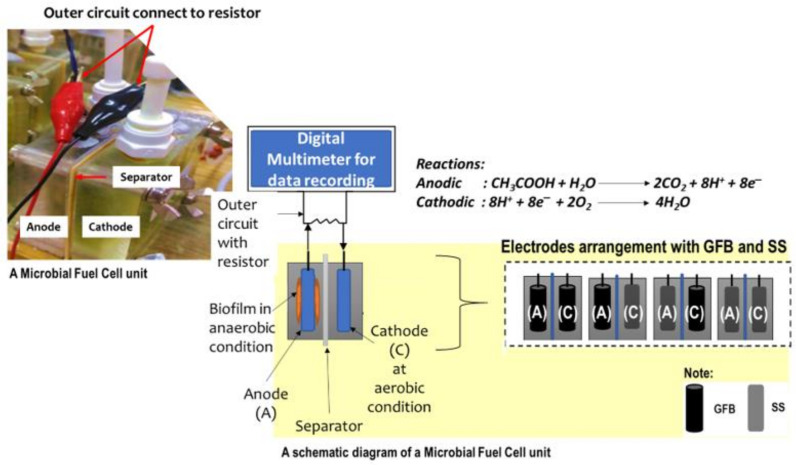
Microbial fuel cell setup.

**Table 1 molecules-27-01045-t001:** Summary of several MFC studies using a biocathode.

Anode/Cathode	Inoculum Anode/Cathode	Power (W/m^2^)	Power (W/m^3^)	Reference
Graphite felts	*Spartina anglica*/Aerobic wastewater	0.24		[30]
Carbon cloth/Activated granular carbon	Anaerobic reactor effluent (ARE)/ARE and aerobic active sludge		1.32	[19]
Carbon brush/Carbon cloth with Pt	Influent from wastewater treatment/Aeration tank	0.29		[14]
Carbon felts	Activated sludge with graphene oxide	0.065		[31]
Carbon papers	Aerobic and anaerobic sludges from municipal wastewater/Aerobic sludge		0.22	[32]
Carbon fibre brushes	Effluent from primary sedimentation tank/Mixture of dewatered sludges: activated tank, digester tank, settling basin and nitrifying tank		7.1	[33]
Ferum/Carbon graphite fibre brushes	Algae/Domestic wastewater		6.6	[34]
Carbon felts	Salt marsh sediment	0.21		[35]

## Data Availability

Not applicable.

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
