# Peer review of "Biotic Cathode of Graphite Fibre Brush for Improved Application in Microbial Fuel Cells"

_molecules, 2022, doi:10.3390/molecules27031045_

Round 1
Reviewer 1 Report
- In the manuscript, it only took “stainless steel” and “graphite fibre brush” as anode and cathode, why not think about “graphite plate” or “graphite rope” to compare with electrode material. Please design your experimental system more carefully.
- Please check the language to make it more concise, and complete references for important conclusions, such as end of the first paragraph in introduction.
- Redraw all figures use different colored lines in the manuscript, enhance the aesthetics and visibility of images, for example, the A, B, C, D insert in Figure 6 is too small, please bring into correspondence with previous graph, and Figure 7 should be drawn more concrete.
- In section of “2.1.2 Biotic Cathode”, OCV should be described in detail, because this is the first time it has appeared in this article. Meanwhile, PBS an COD should be described in detail, too.
- The article mentioned that “Vihodceva et al. obtained raised and flat surfaces after heat-treating the SS at temperatures above 1200 ℃ for 10 min, where the raised surface consisted mainly of nontoxic hematite”. “All COD removal recorded more than 90%, likely contributed by the exoelectrogen and non-exoelectrogen microbes in the anodic chamber. The GFB has a rougher surface than the SS that attracted different colonisation of microbe on the electrode surface and produced high power density”.
Previous studies have shown that hematite is a semiconductor mineral that can significantly enhance the extracellular electron transfer of electroactive microorganisms. It is written in your manuscript that when the SS is heated at 1200℃ for 10 minutes, hematite will be produced on the surface, so hematite will promote microbial electricity generation. How can you confirm that it is not because the presence of hematite increases the power of the MFC?
- Please check and correct symbols in the manuscript. Such as the “℃” symbol in line 155 is wrong.
- In section of discussion say that “In this study, the power generated was much lower than other MFC systems reported with ferricyanide as a catholyte.”, can you summarize the previous study in a table to compared with your result. (who did what and in what year and what was the result and advantages and disadvantages).
- Have repeated experiments or batch experiments been conducted? Can a simple experiment exclude the reliability of data? Thus, the result of polarization curves should have an error margin.
- “The calculation of polarisation curves was done using a potentiostat (Autolab PGSTAT128N, Nederlands). This analysis was carried out below the frequency range of 100 kHz up to 5 mHz, the amplitude AC was 10 mV, and the open-circuit voltage (OCV) was taken once the system was stable. Meanwhile, the internal re-sistance (Rint) was calculated from the slope of the polarisation curve”. Please describe the specific method of measuring polarization curves using a potentiostat, and why measure in the 100 kHz up to 5 mHz range.
- Why not consider about electrochemical impedance to explore the interaction between electrode and biofilm.
- Please specify your SEM model, production place and image acquisition process, and the SEM in figure 6 showed clean and biofilm in graphite brush and stainless steel respectively, but where the sample from, anode or cathode?
Reviewer 2 Report
The paper titled “Biotic-cathode of Graphite Fiber Brush for an Improvement Applications in Microbial Fuel Cell” investigated the effect of the electrode combination between the stainless steel (SS) and graphite fiber brush (GFB) without any chemical modification for constructing bio-electrodes in MFC. The experiments seem to be conducted well, however, I believe the paper does not meet the quality of this journal and recommend submitting the paper to other specific journals. Several comments are given below:
[1] It is hard to find a special novelty in the concept and result of this study. Many previous papers dealt with this topic for increasing the power density using GFB electrodes.
[2] There are many typos, and the English expression must be improved. Many sentences are unclear.
[3] The performance would be time-dependent, the authors can add a plot for time vs. power density in the case of ‘GFBǁGFB’.
[4] The writing flow seems to be not logical in the Introduction part and the paper lacks an explanation for the novelty of this paper.
[5] Why did the authors use HCl solution treatment and annealing at 600 oC for SS pre-treatment?
[6] Description of Preparation of Microbial Fuel Cell Reactors is not clear. What is the material of the Cation exchange membrane? Please provide detailed information of GFB and SS mech including the material, dimension, and the supply company.
[7] Is there any effect of functional groups on the surface of SS, GFB after pre-treatment on the device performance of MFC? I recommend authors conduct XPS measurements to check it.
[8] Provide half and overall reaction of the MFC, and components in the anode/cathode in Figure 7.
Reviewer 3 Report
The authors have submitted a good manuscript providing novel insights about the topic. The abstract is well written, highlighting the methodology and key findings of the reported work. All the sections are well structured, and the study results are supported with proper citations. I recommended the publication with a minor suggestion to the authors.
- The axis titles and legends of figure 1 are very light in color and can be improved. A suggestion is to update the figure with the appropriate color for the axis titles and figure legends. Also, check the same for similar figures to improve the readability.
Round 2
Reviewer 1 Report
- In the manuscript, the numbers on all the coordinate axes of the pictures appear to be blurred, and some lines are of different thickness, please correct them. Due to the large difference in line scale, it is suggested that some diagrams can be further enlarged by embedding method, so as to facilitate readers to understand. You should see pictures from other good papers to improve your picture quality.
- In your coverletter, Question: “Please check the language to make it more concise, and complete references for important conclusions, such as end of the first paragraph in introduction.”; answer: “We have made improvement accordingly: Page 2, Line 16-23.” So why not check all language in manuscript and which article support your view “The concept of MFCs for electricity generation using wastewater, has already been established in research.” at end of the first paragraph in introduction.
- The result is too simple in the manuscript. Please describe your data in detail by comparison.
- The label of Figure 6 (a), (b), (c), (d) takes up more space, please reduce the white background. Some words in figure 7 are compressed, please retain the original shape.
- In your coverletter, “Each MFC sets (treatment and control) had their replications done throughout the experiment. The error margins were reflected in the text instead of the figures under section 2.1, i.e The GFBǁGFB system generated the highest power density, up to 3.03 r 0.7 W/m3 (177A/m3)” Why not show the error margin in the diagram so that the reader can see the data more intuitively. Please add error bars to the figure to improve your manuscript.
- The article mentioned that “Vihodceva et al. obtained raised and flat surfaces after heat-treating the SS at temperatures above 1200 °C for 10 min, where the raised surface consisted mainly of nontoxic hematite”. “All COD removal recorded more than 90%, likely contributed by the exoelectrogen and non-exoelectrogen microbes in the anodic chamber. The GFB has a rougher surface than the SS that attracted different colonisation of microbe on the electrode surface and produced high power density”. Previous studies have shown that hematite is a semiconducting mineral that can significantly enhance the extracellular electron transfer of electroactive microorganisms. It is written in your manuscript that when the SS is heated at 1200°C for 10 minutes, hematite will be produced on the surface, so hematite will promote microbial electricity generation. How can you confirm that it is not because the presence of hematite increases the power of the MFC?
Please clarify my question: Studies have shown that hematite promotes extracellular electron transfer in microorganisms, your experiment produced hematite (Vihodceva et al. obtained raised and flat surfaces after heat-treating the SS at temperatures above 1200 °C for 10 min, where the raised surface consisted mainly of nontoxic hematite). How can you be sure it's not hematite that promotes the extracellular electron transport of microorganisms?
- Please check and correct symbols in the manuscript. Such as the “°C” symbol in line 155 is wrong. Please read my question carefully. the “°C” symbol is wrong in line 12 of 8/15.
- In page 9, “In fact, ferricyanide and other chemical substances with high electron affinity can give higher power performance than the biocathode (Figure 4C)”. How did you make the comparison from Figure 4C?
Reviewer 2 Report
The authors have addressed well the comments raised in the revised manuscript, so I recommend accepting as it is.
Author Response
We thank you Reviewer 2 for your comments to improve the manuscript.